# Control Algorithm Design of a Force-Balance Accelerometer

**DOI:** 10.3390/s23208640

**Published:** 2023-10-23

**Authors:** Zhiqiang Liu, Lei Xia, Bin Wu, Ronghua Huan, Zhilong Huang

**Affiliations:** 1Key Laboratory of Soft Machines and Smart Devices of Zhejiang Province, Department of Mechanics, Zhejiang University, Hangzhou 310027, China; zqliu_w@zju.edu.cn (Z.L.); 11724009@zju.edu.cn (L.X.); zlhuang@zju.edu.cn (Z.H.); 2Advanced Technology Research Institute, Zhejiang University, Hangzhou 310000, China; wubinzju@zju.edu.cn

**Keywords:** time delay, vibration response, optimal control, force-balance sensing

## Abstract

The force-balanced accelerometer (FBA), unlike other types of sensors, incorporates a closed-loop control. The efficacy of the system is contingent not solely on the hardware, but more critically on the formulation of the control algorithm. Conventional control strategies are usually designed for the purpose of response minimization of the sensitive elements, which limits the measurement accuracy and applicable frequency bandwidth of FBAs. In this paper, based on the model predictive control (MPC), a control algorithm of a force-balance accelerometer considering time delay is designed. The variable augmentation method is proposed to convert the force-balance control into an easy-handed measurement error minimization control problem. The discretization method is applied to deal with the time delay problem in the closed loop. The control algorithm is integrated into a practical FBA. The effectiveness of the proposed control is demonstrated through experiments conducted in an ultra-quiet chamber, as well as simulations. The results show that the closed loop in the FBA has a time delay 10 times of the control period, and, utilizing the proposed control, the acceleration signals can be accurately measured with a frequency range larger than 500 Hz. Meanwhile, the vibration response of the sensitive element of the controlled FBA is maintained at the level of microns, which guarantees a large measurement range of the FBA.

## 1. Introduction

Acceleration sensors are instrumental in a myriad of scientific and engineering endeavors. The evolution from the rudimentary open-loop accelerometer to the sophisticated force-balance accelerometers, characterized by their superior linearity, precision, and expansive dynamic range, is noteworthy. Unlike their open-loop counterparts, force-balance accelerometers employ feedback control to counterbalance the input inertial force, thereby quantifying the magnitude of the inertial force to facilitate acceleration measurement [1,2]. In 1990, Rudolf et al. pioneered the design of a force-balance accelerometer specifically for microgravity measurements in spacecraft, achieving an impressive resolution of less than 1 ug at a 1 Hz bandwidth signal [3]. Subsequently, Che et al. developed a force-balance accelerometer characterized by a low sensitivity threshold and high thermal stability, with a sensitivity threshold of $4.6\times 10^{-5}\;g$ for static tests and a range of ±24 g [4]. The advent of micro-electro-mechanical systems (MEMS) in recent years has seen the application of the closed-loop force-balanced control concept to MEMS accelerometers. Liu et al., leveraging electromagnetic force feedback in force-balanced MEMS accelerometers, optimized the circuit structure to compensate for system temperature drift, thereby maintaining the temperature coefficient of the scale factor at a constant 6 ppm/°C [5]. Qu et al., utilizing in-plane currents in MEMS structures to generate spatial magnetic fields that interact with external magnets, achieved an exceptional dynamic range of 157 dB within a 30 Hz bandwidth through electromagnetic force balancing of inertial forces [6].

From the stability of the original classical control to the optimization process of modern control, the robustness threshold problem of most systems against external disturbances is inevitable [7,8,9]. Among them, the time delay problem is particularly prominent [10,11,12,13,14,15], which is often related to the sampling rate of the system itself. In modern control, control problems involving optimal control strategies usually have little robustness to system time delays. Even the most currently used PID control is much less effective in the face of large pure time delays (delay greater than 0.5 times the inertia time). As early as 1958, OJM Smith proposed a time lag prediction compensation algorithm, which was called Smith predictive control. However, Smith’s predictive control requires very precise knowledge of the transfer function between the system input and the controller, which is an impossible task for system identification. Cai et al. proposed an optimal feedback control strategy based on discretizing the traditional continuous state-space equations to solve the linear system time delay to achieve system stability [16,17]. He et al. introduced the free power matrix in the Lyapunov–Krasovskii generalized model transformation to overcome the conservatism caused by the fixed power matrix [18]. Chen et al. designed a targeted disturbance observer to improve the disturbance suppression and performance robustness of the system with time delay [19]. Jin et al. proposed a TDC strategy to enhance the stability robustness and performance robustness of the system based on the traditional time delay control (TDC) approach to significantly improve the position-tracking performance of the servo system [20]. Chen et al. proposed an adaptive robust control based on radial basis function (RBF) to deal with the communication time delay in bilateral nonlinear servo systems. Appropriate neural network parameters enable accurate real-time position capture as well as ambient force feedback [21]. As can be seen, there have been a large number of research results in the area of time delay system stability. These approaches focus on minimizing the system response through direct or indirect compensation for the time delay; however, this is different from the force balance control. Force-balance closed-loop control is the direct offset of external excitation through feedback control forces while minimizing the control system response. Therefore, it not only considers the system stability problem but also pursues the input minimization problem, which is a double guarantee.

In this paper, an MPC-based force-balance control of an acceleration sensing system with time delay is proposed based on the variable augmentation and discretization method. The control method is applied to a practical quartz-flexible FBA. Both simulations and experiments are carried out to validate the effectiveness of the proposed control. The rest of the paper is organized as follows. In Section 2, we introduce the dynamic model of the force-balance acceleration sensing system. The MPC-based force-balance control is obtained in Section 3. In Section 4, the proposed control is applied to a quartz-flexible FBA, and both simulations and experiments are carried out. Section 5 makes a brief conclusion.

## 2. Dynamical Modeling

The force-balance acceleration sensor usually contains a sensitive element and a closed-loop loop. During operation, the sensitive element (quartz flexure beam in Figure 1a) will vibrate under the action of external inertial force caused by the input acceleration signals. The controller computes the feedback control force predicated on the vibration displacement of the sensitive structure, with the primary objective of force balancing being to mitigate the vibration displacement of the sensitive structure. This is achieved by counterbalancing the external excitation with the feedback control force. In this scenario, the control force is equal and opposite to the external excitation. However, despite the inherent advantages of closed-loop control systems in achieving precision, they are typically marked by an intrinsic temporal delay. As depicted in Figure 1b, the feedback control executed at a given instant is predicated on the displacement observed at a preceding moment (t−τ). This mechanism does not assure equilibrium within the system, much less facilitate minimization of vibrations within sensitive structures. Consequently, addressing this temporal lag is of paramount importance for any system striving for precision control. This aspect has been thoroughly explored and considered in our study.

For a multi-axial force-balance accelerometer, the motion equation of the sensitive elements has the form
(1)$$M\ddot{X}\left(t\right)+C\dot{X}\left(t\right)+KX\left(t\right)=F\left(t\right)-U\left(t-\tau \right)$$
where X is the vibration displacements of the sensitive elements. ***M***, ***C***, and ***K*** are the mass, damping, and stiffness matrix of the sensitive elements, respectively. $F\left(t\right)$ is the inertial force excitation vector due to the input of acceleration. $U\left(t-\tau \right)$ is the feedback control force vector. τ is the time delay in the whole closed-loop system.

System (1) can be rewritten in the following state-space form:(2)$${\dot{\alpha }}_{1}=A_{1}\alpha_{1}+B_{1}\left[F\left(t\right)-U\left(t-\tau \right)\right]$$
where $\alpha_{1}={\left[X\left(t\right),\dot{X}\left(t\right)\right]}^{T}$ and
$$A_{1}=\left[\begin{array}{cc} 0 & I\\ -M^{-1}K & -M^{-1}C \end{array}\right],\;B_{1}=\left[\begin{array}{c} 0\\ M^{-1} \end{array}\right]$$

The main purpose of the force-balance control is to design a control force (U) to balance the external inertial force (F) to realize the measurement of the acceleration signal. Differing from the traditional response minimization control [22,23,24], in the force-balance sensing system, the excitations are unknown and need to be measured, which makes the control force difficult to determine. To solve this problem, the present authors have proposed a linear quadratic regulator (LQR) control strategy of a force-balance sensing system for measurement error minimization [25]. The effectiveness of this control strategy has been verified by simulations. The approach of formulating state-space equations with measurement errors serving as state vectors offers a theoretical foundation for devising optimal control strategies. However, this remains largely theoretical and lacks empirical validation, particularly in the context of system time lag disturbances. In this paper, we extend this transformational framework to address system time lags and compute feedback control based on MPC.

## 3. Feedback Control Design

The purpose of force-balance control is to balance the control force and external inertial force, that is, to make the difference between the control force and external inertial force minimal. Since the inertial force and control force are both unknown, it is difficult to obtain the control force directly. To solve this problem, the procedure necessitates two expansions of the state-space equation, with an intermediate step involving the discretization of a continuous state-space equation.

First, we introduce the difference between the control force and the external inertia force as a new variable into the system for the first augmentation of the state vector, i.e.,
(3)$$\mu_{1}\left(t-\tau \right)=F\left(t\right)-U\left(t-\tau \right)$$
where $\mu_{1}$ denotes the measurement error. The first-order derivative of $\mu_{1}$ is given as
(4)$$U_{2}(t-\tau )={\dot{\mu }}_{1}\left(t-\tau \right)=\dot{F}(t)-\dot{U}\left(t-\tau \right)$$

Then, the state-space Equation (2) is extended as
(5)$${\dot{\alpha }}_{2}=A_{2}\alpha_{2}+B_{2}U_{2}\left(t-\tau \right)$$
where $\alpha_{2}={\left[X\left(t\right),\dot{X}\left(t\right),\mu_{1}\left(t-\tau \right)\right]}^{T}$ is the extended state vector, and the extended state matrix ($A_{2}$) and input matrix ($B_{2}$) are given as follows:$$A_{2}=\left[\begin{array}{ccc} 0 & I & 0\\ -M^{-1}K & -M^{-1}C & M^{-1}\\ 0 & 0 & 0 \end{array}\right],\;B_{2}=\left[\begin{array}{c} 0\\ 0\\ I \end{array}\right]$$

$U_{2}$ can be regarded as the equivalent control force of the new controlled system (5). It can be seen that introducing the measurement error ($\mu_{1}$), the original force-balance control problem (2) is converted into a traditional response minimization control (5). Once $U_{2}$ is determined, the force-balance control force (U) can be obtained according to Formulas (3) and (4).

The crux of the matter lies in the fact that the feedback control force, as depicted in Equation (5), incorporates a time delay. Consequently, it is not feasible to directly derive the present control force via an MPC solution. To obtain the optimal control force, the system (5) is first discretized in time.

Before that, it is necessary to discuss the classification of the time lag (τ), which is divided into two cases:
(1)$\tau =lT_{0}$;(2)$\tau =lT_{0}-\epsilon$, $0<\epsilon <T_{0}$ (l is a positive integer).

Cases 1 and 2 indicate that the time lag is an integer and non-integer multiple of the control period, respectively. Based on the categorization of the above two cases, the transformation of Equation (5) into discrete form is also categorized into two types (see Appendix A for details):(6a)$$\begin{array}{r} \alpha_{2}\left(k+1\right)=e^{A_{2}T_{0}}\alpha_{2}\left(k\right)+\int_{0}^{T_{0}}e^{A_{2}\lambda }B_{2}d\lambda \hat{U}\left(\;k-l\right)\\ +\int_{0}^{T_{0}}e^{A_{2}\lambda }B_{2}\hat{F}\left(\;kT_{0}+T_{0}-\lambda \right)d\lambda \end{array}$$
for case 1, and
(6b)$$\begin{array}{c} \alpha_{2}\left(k+1\right)=e^{A_{2}T_{0}}\alpha_{2}\left(k\right)+\int_{0}^{\epsilon }e^{A_{2}\lambda }B_{2}d\lambda \hat{U}\left(\;k-l+1\right)\\ \;\;\;\;\;\;\;\;\;\;\;\;\;\;+\int_{\epsilon }^{T_{0}}e^{A_{2}\lambda }B_{2}d\lambda \hat{U}\left(\;k-l\right)+\int_{0}^{T_{0}}e^{A_{2}\lambda }B_{2}\hat{F}\left(\;kT_{0}+T_{0}-\lambda \right)d\lambda \end{array}$$
for case 2. $\hat{U}=-\dot{U}$, $\hat{F}=\dot{F}$.

Equations (6a) and (6b) can be rewritten as
(7a)$$\alpha_{2}\left(k+1\right)=\hat{A}\alpha_{2}\left(k\right)+\hat{B}\hat{U}\left(k-l\right)+W(k)$$
and
(7b)$$\alpha_{2}\left(k+1\right)=\hat{A}\alpha_{2}\left(k\right)+\hat{B^{\prime }}\hat{U}\left(\;k-l\right)+\hat{C}\hat{U}\left(\;k-l+1\right)+W\left(k\right)$$
respectively, where
$${\hat{A}=e}^{A_{2}T_{0}},\;\;\hat{B}=\int_{0}^{T_{0}}e^{A_{2}\lambda }B_{2}d\lambda ,$$
$$\hat{B^{\prime }}=\int_{\epsilon }^{T_{0}}e^{A_{2}\lambda }B_{2}d\lambda ,\;\hat{C}=\int_{0}^{\epsilon }e^{A_{2}\lambda }B_{2}d\lambda ,\;W(k)=\int_{0}^{T_{0}}e^{A_{2}\lambda }B_{2}\hat{F}\left(\;kT_{0}+T_{0}-\lambda \right)d\lambda$$

Introducing $\hat{U}\left(\;k-1\right)$,$\;\hat{U}\left(\;k-2\right)$,$\cdots \;\hat{U}\left(\;k-l\right)$ as new state variables for the second augmentation of the state vector, Equations (7a) and (7b) can be extended as
(8)$$Z(k+1)=EZ\left(k\right)+G\hat{U}\left(k\right)+DW(k)$$
where
$$Z\left(k\right)=\left[\begin{array}{c} \alpha_{2}\left(k\right)\\ \hat{U}\left(\;k-l\right)\\ \begin{array}{c} \hat{U}\left(\;k-l+1\right)\\ \begin{array}{c} \vdots \\ \hat{U}\left(\;k-1\right) \end{array} \end{array} \end{array}\right]\;G=\left[\begin{array}{c} 0\\ 0\\ \begin{array}{c} \vdots \\ \vdots \\ I \end{array} \end{array}\right]\;D=\left[\begin{array}{c} I\\ 0\\ \begin{array}{c} \vdots \\ \vdots \\ 0 \end{array} \end{array}\right]$$
$${E=E}_{1}=\left[\begin{array}{ccc} \hat{A} & \begin{array}{cc} \hat{B} & \begin{array}{cc} 0 & \cdots \end{array} \end{array} & 0\\ \begin{array}{c} 0\\ \vdots \\ \vdots \end{array} & \begin{array}{ccc} 0 & I & \ddots \\ 0 & 0 & \ddots \\ 0 & 0 & \ddots \end{array} & \begin{array}{c} \vdots \\ \vdots \\ I \end{array}\\ 0 & \begin{array}{cc} \ldots & \ldots \end{array} & 0 \end{array}\right]\;(\mathrm{case}\;1)$$
$$E=E_{2}=\left[\begin{array}{ccc} \hat{A} & \begin{array}{cc} \hat{B} & \begin{array}{cc} \hat{C} & \cdots \end{array} \end{array} & 0\\ \begin{array}{c} 0\\ \vdots \\ \vdots \end{array} & \begin{array}{ccc} 0 & I & \ddots \\ 0 & 0 & \ddots \\ 0 & 0 & \ddots \end{array} & \begin{array}{c} \vdots \\ \vdots \\ I \end{array}\\ 0 & \begin{array}{cc} \ldots & \ldots \end{array} & 0 \end{array}\right]\;(\mathrm{case}\;2)$$

Following discretization, the initial time-delay control issue (5) is converted into a control problem (8) devoid of explicit time delay. Despite the fact that this transformation results in an augmentation of the state vector’s dimensionality, optimal control forces can still be readily derived for linear systems.

To derive the feedback control force for the aforementioned optimal control problem (8), this study employs MPC for computation. Before this, it is necessary to establish a hypothesis that aligns with practical implications. In practical engineering applications, it is necessary to select the sampling rate according to the frequency band of the vibration acceleration signal. In order to guarantee the measurement effect of the sensor, the sampling frequency is usually required to be much higher than the frequency band of the acceleration signal. This means that in a sampling or control period ($T_{0}$), the acceleration signal or inertial force ($F\left(t\right)$) can be regarded as a slowly varying variable, i.e., $\dot{F}\left(t\right)≃0$. Then, Equation (8) can be simplified as
(9)$$Z(k+1)=EZ\left(k\right)+G\hat{U}\left(\;k\right)$$

For MPC control, we have a rolling forecast space. Assuming a forecast dimension of N, the entire forecast equation can be expressed as
(10)$$\begin{array}{l} Z\left(k|k\right)\;=\;Z\left(k\right)\\ Z\left(k+1|k\right)=EZ\left(k\right)+G\hat{U}\left(k|k\right)\\ Z\left(k+2|k\right)=E^{2}Z\left(k\right)+EG\hat{U}\left(k|k\right)+G\hat{U}\left(k+1|k\right)\\ \;\;\;\;\;\;\;\;\;\;\;\;:\\ \;\;\;\;\;\;\;\;\;\;\;\;:\\ Z\left(k+N|k\right)=E^{N}Z\left(k\right)+E^{N-1}G\hat{U}\left(k|k\right)+\ldots +G\hat{U}\left(k+N-1|k\right) \end{array}$$
collated to obtain
(11)$$\bar{Z}=SZ\left(k\right)+H\bar{U}$$
where
(12)$$\begin{array}{c} \bar{Z}=\left[\begin{array}{c} Z\left(k|k\right)\;\\ Z\left(k+1|k\right)\\ \begin{array}{c} Z\left(k+2|k\right)\\ \;:\\ \begin{array}{c} \;:\\ Z\left(k+N|k\right)\; \end{array} \end{array} \end{array}\right]\;\;\;\;\;\;\;\;\;\;\;\;\;\;\;\;\;\bar{U}=\left[\begin{array}{c} \hat{U}\left(k|k\right)\;\\ \hat{U}\left(k+1|k\right)\\ \begin{array}{c} \hat{U}\left(k+2|k\right)\\ \;:\\ \begin{array}{c} \;:\\ \hat{U}\left(k+N-1|k\right)\; \end{array} \end{array} \end{array}\right]\\ S=\left[\begin{array}{c} I\;\\ E\\ \begin{array}{c} E^{2}\\ \;:\\ \begin{array}{c} \;:\\ E^{N}\; \end{array} \end{array} \end{array}\right]\;\;\;\;\;\;\;\;\;\;\;\;\;\;H=\left[\begin{array}{ccc} \begin{array}{cc} \begin{array}{cc} 0 & \end{array} & \;\;\;\;0 \end{array} & .. & 0\\ \begin{array}{cc} \begin{array}{cc} G & \end{array} & \;\;\;\;0 \end{array} & .. & 0\\ \begin{array}{cc} \begin{array}{c} EG\\ :\\ \begin{array}{c} :\\ E^{N-1}G \end{array} \end{array} & \begin{array}{c} G\\ :\\ \begin{array}{c} :\\ E^{N-2}G \end{array} \end{array} \end{array} & \begin{array}{c} ..\\ :\\ \begin{array}{c} :\\ .. \end{array} \end{array} & \begin{array}{c} 0\\ :\\ \begin{array}{c} :\\ G \end{array} \end{array} \end{array}\right] \end{array}$$

For force-balance sensing control, the control objectives mainly include the following: (1) minimum measurement error to ensure the sensing accuracy; (2) the displacement of the sensitive element is small, which ensures a large measurement range of the sensor; (3) the output control force should not be too large to reduce the energy consumption of the sensor. According to these requirements, the performance index of MPC control can be designed as follows:(13)$$\begin{array}{c} J=\sum_{i=0}^{N-1}Z{\left(k+i|k\right)}^{T}QZ\left(k+i|k\right)+\hat{U}{\left(k+i|k\right)}^{T}R\hat{U}\left(k+i|k\right)\\ +{Z\left(k+N|k\right)}^{T}Q_{F}Z\left(k+N|k\right) \end{array}$$
where Q, R, and $Q_{F}$ are the weight matrices. In the case where Q, R, and $Q_{F}$ are all diagonal matrices, Equation (13) is simplified as
(14)$$J=\sum_{i=0}^{N}||{\Gamma_{z,i}Z\left(k+1\right)||}^{2}+\sum_{i=0}^{N-1}{\left||\Gamma_{\hat{u},i}\hat{U}\left(k+1\right)|\right|}^{2}$$
where
$$\Gamma_{z}≝\sqrt{diag\left(Q,Q,\ldots Q,Q_{F}\right)}$$
$$\Gamma_{\hat{u}}≝\sqrt{diag\left(R,R,\ldots R,R\right)}$$

The purpose of MPC is to minimize the index, i.e.,
(15)$${min}_{\;\hat{U}}\;J\left(Z,\hat{U},N\right)$$

Then, the transformation is introduced:(16)$$\rho ≝\left[\begin{array}{c} \Gamma_{z}\bar{Z}\\ \Gamma_{\hat{u}}\bar{U} \end{array}\right]$$

Then, Equation (14) can be rewritten as
(17)$$J\left(Z,\hat{U},N\right)=\rho^{T}\rho$$

Substituting Equation (11) into Equation (16), Equation (16) becomes
(18)$$\rho =\left[\begin{array}{c} \Gamma_{z}SZ+\Gamma_{z}H\bar{U}\\ \Gamma_{\hat{u}}\bar{U} \end{array}\right]=\left[\begin{array}{c} \Gamma_{z}H\\ \Gamma_{\hat{u}} \end{array}\right]\bar{U}+\left[\begin{array}{c} \Gamma_{z}SZ\\ 0 \end{array}\right]=A_{3}\bar{U}+B_{3}$$
where
(19)$$A_{3}=\left[\begin{array}{c} \Gamma_{z}H\\ \Gamma_{\hat{u}} \end{array}\right]\;B_{3}=\left[\begin{array}{c} \Gamma_{z}SZ\\ 0 \end{array}\right]$$

As we can see,
(20)$${min}_{\;\hat{U}}\;J={min}_{\;\hat{U}}\;\rho^{T}\rho \;\;$$

The MPC control can be obtained by taking the minimum value on the right side of Equation (20), i.e.,
(21)$$\frac{d\rho^{T}\rho }{d\bar{U}}=2{A_{3}}^{T}(A_{3}\bar{U}+B_{3})=0$$

Then, the MPC control force is determined:(22)$$\bar{U}=-({A_{3}}^{T}A_{3}{)}^{-1}{A_{3}}^{T}B_{3}\;\;$$

Since $\hat{U}=-\dot{U}$, the original force-balance control $U\left(k\right)$ is finally obtained:(23)$$\left(k\right)=\dot{U}\left(k\right)T_{0}+U\left(k-1\right)=-\hat{U}\left(k\right)T_{0}+U\left(k-1\right)$$
where $\hat{U}\left(k\right)$ is the first term in the vector $\bar{U}$ (shown in Equation (12)) obtained from Equation (22).

## 4. Application to a Quartz Flexible FBA

The proposed control strategy is applied to a practical quartz-flexible force-balance accelerometer. The structure and control circuit diagram of the acceleration sensor is shown in Figure 2a. The dark blue rectangular structure in the middle of the structure is a flexible quartz beam, which is the sensitive element of the sensor and also the receptor of external inertial excitation. When the sensitive beam vibrates, the yellow capacitive pole plate at its end detects the corresponding displacement. The capacitance data obtained using AD sampling are input to the FPGA for the calculation of the control output, and the output current is fed to the red solenoid coil to generate the magnetic torque feedback to the sensitive beam to achieve the force balance. Figure 2b shows the test devices and Figure 2c shows the debug interface and the integrated circuit board. Figure 2d is the package prototype of the quartz-flexible accelerometer.

The sensitive beam can be modeled as a second-order vibrating system. The transfer function of the entire system needs to be identified first. The magnitude of each parameter of the transfer function (including the time lag) is obtained by the indirect identification method, which performs an original rational analysis in terms of the response of a low-order vibrating system [26]. The open-loop transfer function under a step excitation is identified as following:(24)$${tf}_{open}=\frac{5.917\times 10^{8}}{s^{2}+1700s+2.495\times 10^{6}}e^{-\frac{1}{1\times 10^{4}}s}$$

It is seen from the transfer function (24) that the whole time delay of the sensing system is 10^−4^ s. In the experiments, the selected sampling frequency and control frequency are 100 kHz. The corresponding control period is $T_{0}=1\times 10^{-5}\;s$. This means the delay time is 10 times the control period, which belongs to the long delay control problem. If the time delay is not considered during the design of the control algorithm, the controlled system will easily lose its stability. According to the method proposed in this paper, the optimal control force can be derived in the form of (21). The elements of the weight matrices (Q) and control parameter (R) in the performance index (13) take the following values:$${Q\;}_{1,1}=2\times 10^{9},\;{Q\;}_{2,2}=0,\;{Q\;}_{3,3}=5\times 10^{7},\;{Q\;}_{4,4}\sim {Q\;}_{13,13}=1;\;Q_{F}=Q;\;R=1.806\times 10^{-6}$$

And the elements unmentioned are 0. The elements ${Q\;}_{1,1}$ and ${Q\;}_{3,3}$ determine the weight of measurement error and displacement of the sensitive beam. This is the reason why they set large values. The presence of a large time delay leads to an expansion of the state vector dimension. For the purpose of good control effectiveness, the prediction space (*N*) is set as a relatively large value of 500.

Figure 3 gives the Bode plot of the optimally controlled FBA (simulation), where the input is the external excitation signal (independent of the type of signal), and the output is the measurement error. The amplitude–frequency curve in the Bode diagram shows the relative error of measurement. It can be seen that the frequency range with the relative measurement error under −10 dB is about 0~1000 Hz. The controlled FBA has a large applicable frequency band. The phase frequency curve maintains a constant value up to 383.1 Hz. In this frequency range, the system maintains a constant phase difference between the measured signal and the external excitation. It is very important for phase compensation in practical sensing applications.

The measurement accuracy of the control method will be further verified through the following numerical simulations and experiments.

### 4.1. Numerical Simulation Results

The closed-loop simulation program is built using SIMULINK, where the sampling time of the zero-order retainer is $T_{0}=1\times 10^{-5}\;s$ and the time delay $\tau =10T_{0}$. A rectangular wave acceleration signal is applied to the sensing system as the external inertial force excitation. The proposed control is executed to balance the inertial force, from which the measured acceleration signal is obtained. Figure 4 displays the input and the measured acceleration signals for rectangular waves with different amplitudes. Obviously, the measured signals match the input signal quite well. The rectangular acceleration signals can be accurately measured by using the proposed control. Figure 5 shows the displacement of the sensitive element under the proposed control. It can be seen that when the rectangular wave just applies to the sensor, the sensitive element will have a large displacement due to the step input. However, the response attenuates rapidly under the control force and finally makes a slight vibration near the equilibrium position. Under the excitation of a 2.5 g acceleration signal, the maximum displacement of the sensing element is only 60 microns, which is much less than the maximum displacement limit (200 microns) of the sensor.

Subsequently, the FBA integrated with the proposed control is used to measure a set of periodic acceleration signals with frequencies of 25 Hz, 200 Hz, and 500 Hz, respectively. Figure 6a–c show the measurement results under 25 Hz, 200 Hz, and 500 Hz periodic acceleration input signals, respectively. Again, the measured results agree well with the actual inputs. For 500 Hz input signals, there are few measurement errors in the peak values in Figure 6c. This implies that as the input frequency increases, the measurement error increases accordingly for a fixed weight matrix (Q) and control parameter (R). However, Q and R can be adjusted appropriately for the practical frequency band requirements. Figure 6d delineates the dynamic displacement of the sensor’s sensitive element, subjected to periodic acceleration excitations with an amplitude of 1g across a spectrum of frequencies. A noteworthy observation is the existence of a marginal phase deviation between the measured signal and the actual input, and this phase deviation is compensated for in Figure 6a–c. This phase deviation remains relatively unaltered up to a frequency threshold of 380 Hz, which is corroborated by the phase–frequency curves in Figure 3. As a consequence, notwithstanding the constancy in the amplitude of the input signal, an escalation in frequency engenders an amplification in the phase difference. This, in turn, culminates in a disparate displacement response output shown in Figure 6d. However, it is unequivocally discernible that the displacement incurred by the sensitive mass is substantially inferior to the maximum displacement capacity of the sensor.

### 4.2. Experimental Results

In this section, we conduct experiments to further validate the proposed control methodology. The experimental procedure is as follows. Initially, digital excitation signals are supplied to the system via software (Xilinx_Vivado 2017.4), which are then converted into current signals via a circuit and fed into an electromagnetic coil. The resultant magnetic field interacts with the inherent magnetic field of a permanent magnet, thereby generating an external excitation on a sensitive beam. Subsequently, differential capacitance is utilized to capture the displacement of the sensitive beam, which is then relayed to an FPGA and processed by a pre-programmed control algorithm to compute control signals. These control signals interact with a fixed magnetic field via an electromagnetic coil to generate a feedback control force that counteracts the excitation on the sensitive structure. This feedback control force serves as a representation of input excitation acceleration.

To avoid environmental disturbances, all tests are conducted in an ultra-quiet room. Figure 7 shows the power spectral density of environmental noise in the ultra-quiet room. It can be clearly seen that, despite being in an ultra-quiet room, the low-frequency noise still exists with the intensity of the mg order. The low-frequency noise may be caused by both the circuit board and environmental interference. But the high-frequency noise has been greatly reduced. The environmental noise will limit the system’s resolution in the order of mg. However, if the specific bandwidth of the signal to be measured is known, data filtering can also be carried out to obtain a better resolution. Figure 8 presents the Allan variance of the zero-output drift of the controlled FBA, which shows that the sensing system has good zero-drift stability.

Some measured results of rectangular wave inputs are given in Figure 9. The experimental results show that the proposed control method can effectively measure the square wave input with different amplitudes. Unlike the numerical results shown in Figure 4, the measured results in the experiments have an initial overshoot. However, the overshoot will quickly fall back to a stable state. The overshoot lies in the following two reasons: one is the inaccuracy of the system identification, and the other is the noise disturbances.

The displacement of the sensitive element is also measured for rectangular wave inputs, as shown in Figure 10. The experimental results are consistent with the simulation results in Figure 5. Similar to the numerical results in Figure 5, the sensitive element has a large displacement at the moment when the rectangular wave signal is applied. However, the response will attenuate quickly to the equilibrium position.

Figure 11a–c display the performance of the sensor for measuring periodic acceleration signals with amplitude 2.5 g and frequencies 25 Hz, 200 Hz, and 500 Hz, respectively. The measured signals coincide with the actual inputs, even for a 500 Hz input. The effectiveness of the proposed control is validated experimentally. However, it is obvious that, for the high-frequency input in Figure 10c, the waveform is somewhat deformed. This is due to the low number of sampling points in the system. The waveform deform can be improved by increasing the digital sampling rate of the FPGA. Figure 11d plots the displacements of the sensitive beam of the sensor for periodic acceleration inputs with different frequencies. Similarly, the displacement of the sensitive beam increases with the increment of the input frequency. For a signal with 200 Hz, the maximum displacement of the sensitive beam is about 5 microns. The very small stable response guarantees a large measurement range of the force-balance sensor.

## 5. Conclusions

In this paper, an MPC-based force-balance control of an acceleration sensor considering time delay was proposed. Both numerical simulations and experiments were carried out to validate the proposed control strategy. The design process of the control strategy includes three key steps: (1) the measurement error is introduced as a new variable so that the difficult force-balance control problem is transformed into the easy-handed measurement error minimization control problem with time delay; (2) the time discretization method is applied to transform the force-balance sensing system with time delay into a system contains no time delay; (3) the expression of the MPC control force is determined.

The proposed control is integrated into a practical quartz-flexible accelerometer. Through system identification, it is found that there is a long time delay, up to 10 times the control period (*T*_0_), in the closed loop of the accelerometer. The proposed control strategy has achieved strong robustness for large time delays. The results show the following: (1). Under a time delay that is 10 times the size of the control period, the system can still maintain very good stability, and the system state is far less than its permissible range. (2). Simulation data shows that under the consideration of long delays, the measurement bandwidth of controlling FBA (force-balanced accelerometer) can reach 1000 Hz, while the experimental measurement bandwidth is up to 500 Hz. (3). According to the experimental results, the output stability of the system is 8.67 × 10^−5^ g under this control algorithm. The control method described in this article not only achieves the measurement of acceleration signals but also effectively suppresses the vibration displacement of the sensor, thereby ensuring the large dynamic measurement range of the sensor.

## Figures and Tables

**Figure 1 sensors-23-08640-f001:**
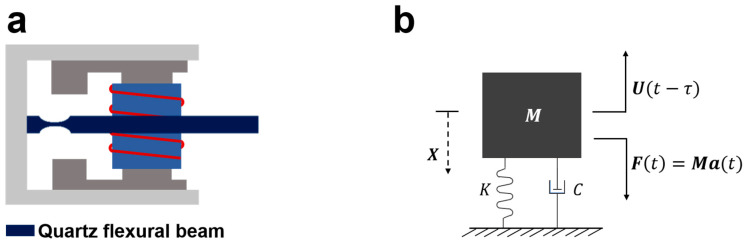
Dynamic modeling of sensitive structures of a uniaxial accelerometer. (**a**) Section of the internal structure of the adapted flexural accelerometer, with the quartz flexural vibrating beam in dark blue; (**b**) equivalent dynamics modeling of the adapted flexural beam.

**Figure 2 sensors-23-08640-f002:**
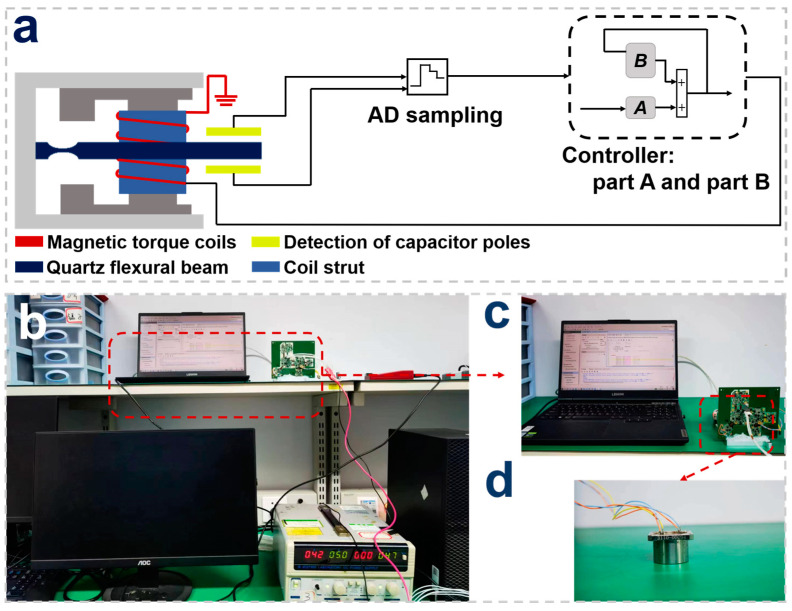
Device and equipment of the experiment. (**a**) The structure and control circuit diagram of the accelerometer. The part A is based on the state of the system and part B control is feedback control from past moments; (**b**) overview of experimental equipment; (**c**) PC operating interface and circuit board; (**d**) prototype of the quartz-flexible accelerometer.

**Figure 3 sensors-23-08640-f003:**
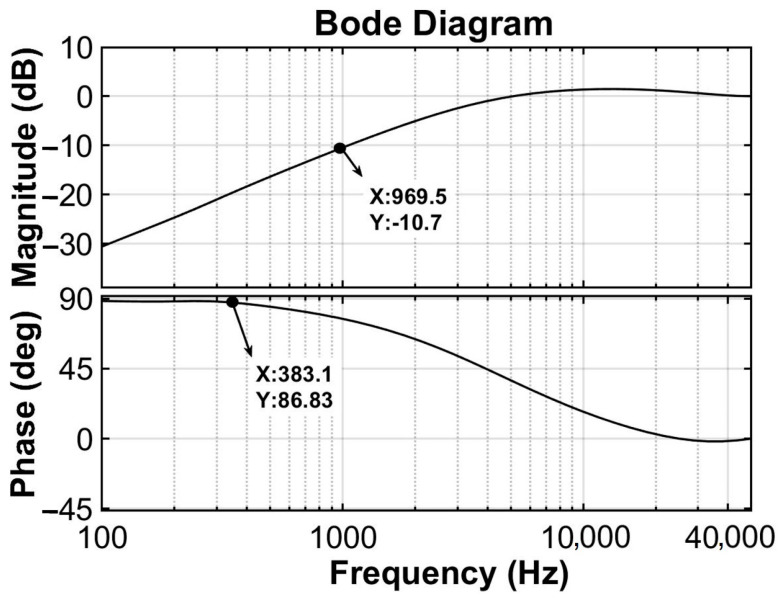
Bode plot of closed-loop measurement error and input.

**Figure 4 sensors-23-08640-f004:**
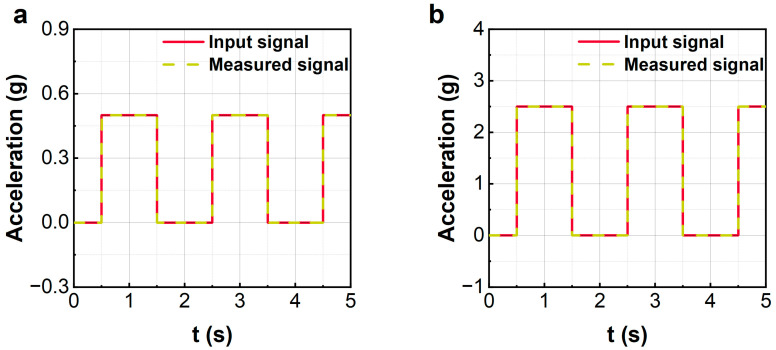
Simulation of rectangular wave signal measurements. (**a**): 0.5 g; (**b**): 2.5 g.

**Figure 5 sensors-23-08640-f005:**
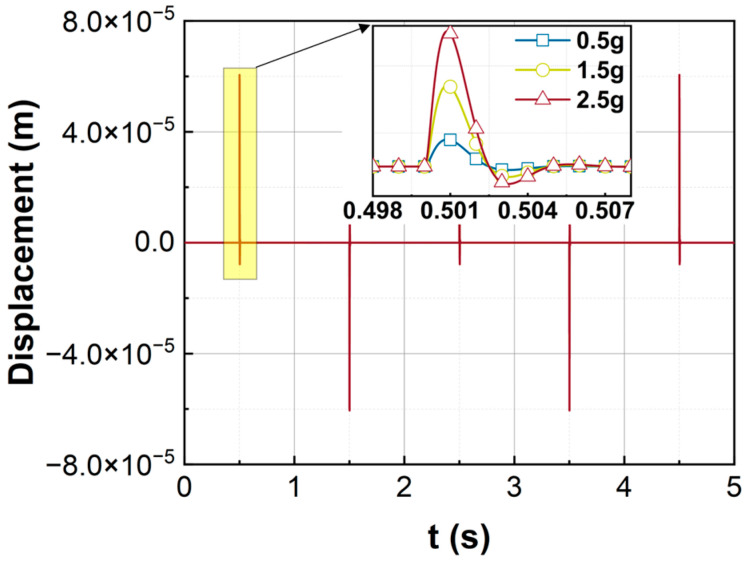
Simulation of the responses for rectangular wave inputs.

**Figure 6 sensors-23-08640-f006:**
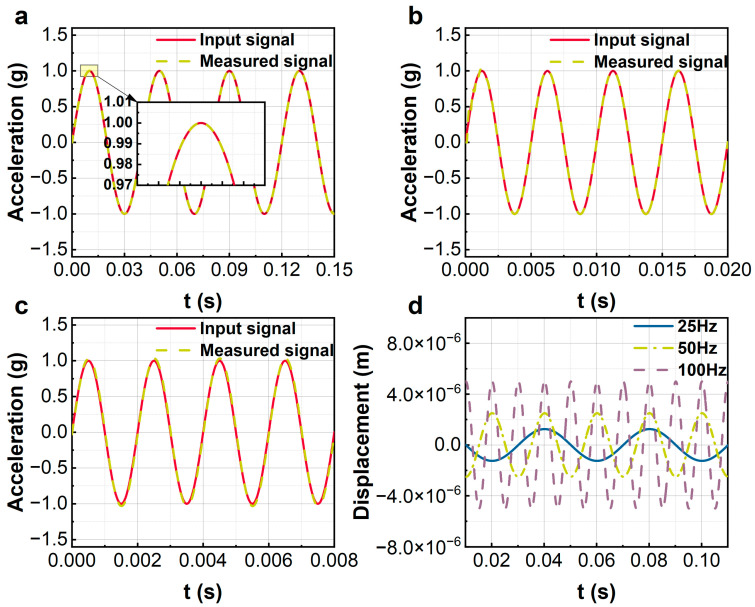
Simulation of sine wave measurements. (**a**–**c**) Input frequencies of 25 Hz, 200 Hz, and 500 Hz; (**d**) the vibration response of the sensitive element.

**Figure 7 sensors-23-08640-f007:**
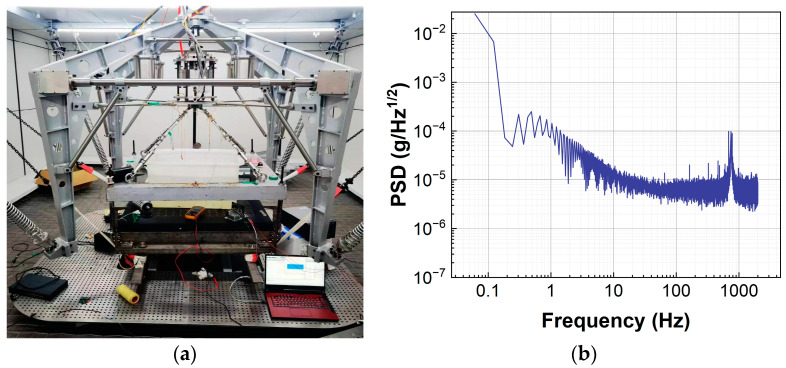
Analysis of the noise of the system. (**a**) Ultra quiet chamber test rig; (**b**) power spectral density of environmental noise.

**Figure 8 sensors-23-08640-f008:**
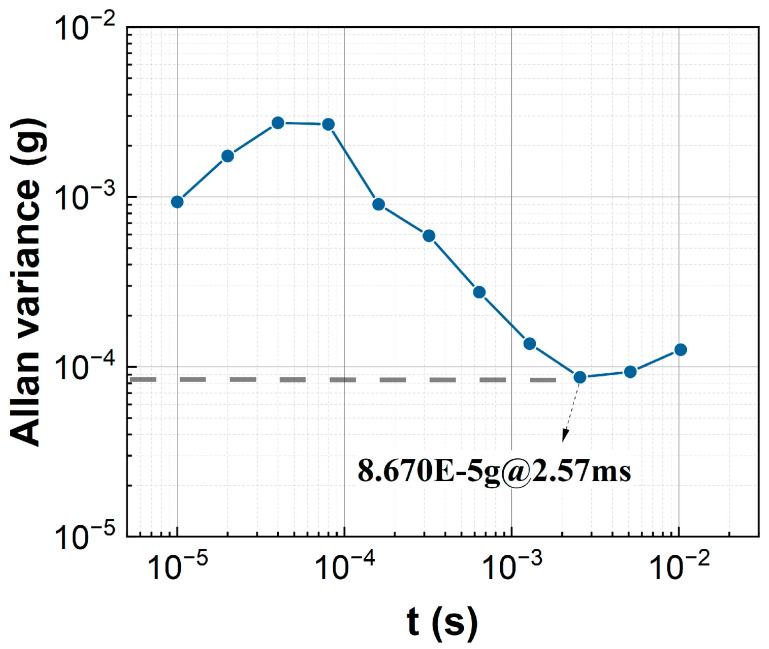
Allan variance of the zero-output drift of the controlled FBA.

**Figure 9 sensors-23-08640-f009:**
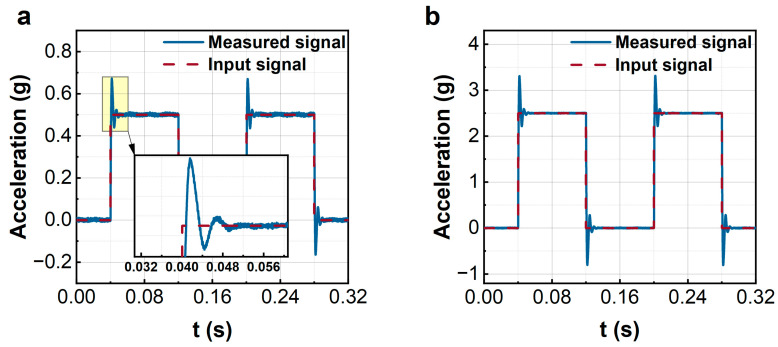
The experiments of rectangular wave signal measurements. (**a**): 0.5 g; (**b**): 2.5 g.

**Figure 10 sensors-23-08640-f010:**
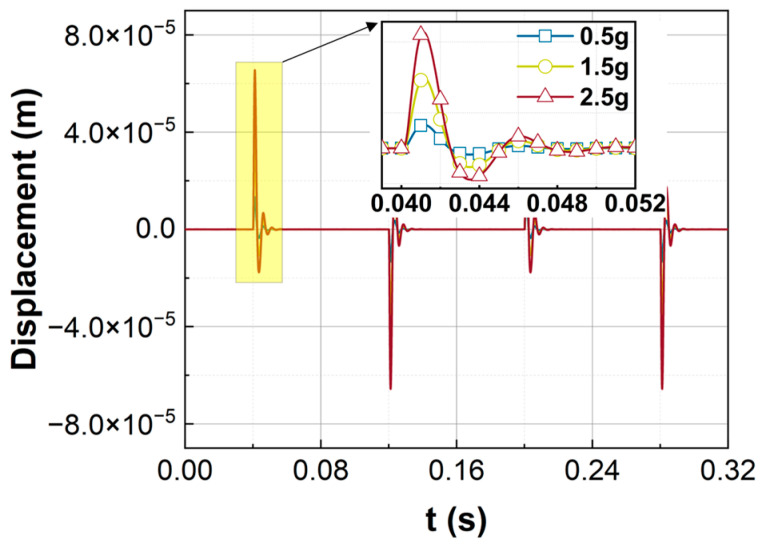
The experiments of the responses for rectangular wave inputs.

**Figure 11 sensors-23-08640-f011:**
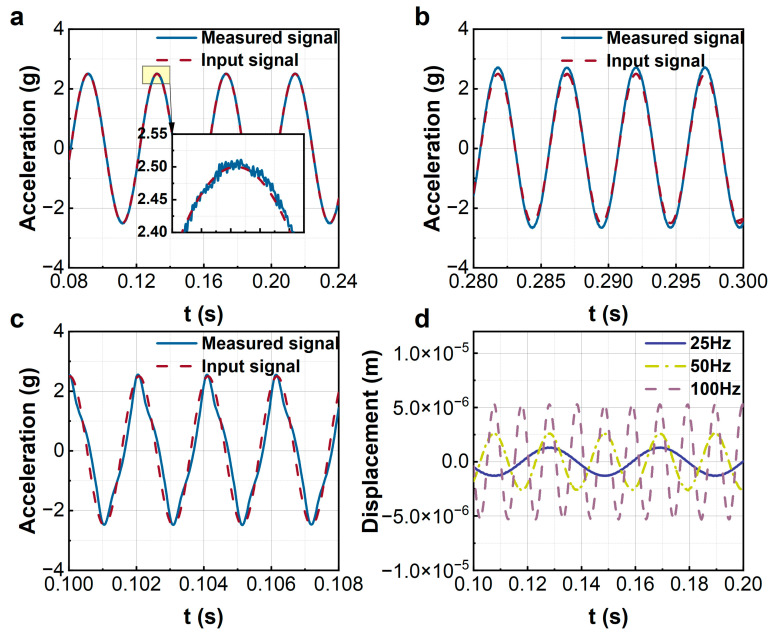
The experiments of sine wave measurements. (**a**–**c**) Input frequencies of 25 Hz, 200 Hz, and 500 Hz; (**d**) the vibration response of the sensitive element.

## Data Availability

The data that support the findings of this study are available from the corresponding author upon reasonable request.

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
