# Peer review of "Control Algorithm Design of a Force-Balance Accelerometer"

_sensors, 2023, doi:10.3390/s23208640_

Round 1
Reviewer 1 Report
Comment:
In this paper, an MPC based control algorithm for force balance accelerometer is designed, which is of research interest in the field of force balance accelerometer research, but there are some problems in the article, and it is recommended that the authors make careful revisions. The following are the specific comments:
Specific Comments:
1. The author published the paper "CONTROL ALGORITHM OF A MEMS FORCE-BALANCE ACCELEROMETER FOR MINIMIZING THE MEASUREMENT ERROR (DOI: 10.6052/1672-6553-2022-002)" in "JOURNAL OF DYNAMICS AND CONTROL" in 2022. That study has already provided a detailed description of the control algorithms involved in this paper, and this paper is a practical application of the algorithms. Therefore, does the new content of this study support the publication of the paper as an "Article"? If it is difficult to support the new content of the study, it is recommended that the type of the paper be changed to "Communication".
2. Line 218 of the text mentions that within the measurement error range of -10 dB, the corresponding frequency range of the accelerometer is 0-1000 Hz. However, according to the displacement plots of the response of the sensitive original of the accelerometer in Fig. 5(d) and Fig. 10(d), it can be seen that under the condition of the same acceleration, the amplitude of the sensitive original varies with the frequency, and it varies with a large magnitude, and the difference of 25 Hz and 100 Hz is over 50%. This is unreasonable, theoretically, the accelerometer in the corresponding range of frequency, the amplitude of the sensitive element is only related to the acceleration, that is, when the acceleration is the same, the amplitude of the movement of the sensitive element should not be very different.
3. From Fig. 10(c), it can be seen that the output signal of the force balance accelerometer based on this control method has been distorted at a frequency of 500 Hz, and the excitation vibration signal cannot be measured accurately. Therefore, does the accelerometer have a frequency response range of 1000 Hz?
Author Response
see the attached files.

Reviewer 2 Report
This paper proposes an MPC-based control algorithm with the variable augmentation method to convert the force-balance control into a response minimization control. Both numerical and experimental investigation is conducted, the research is within the scope of this paper. At current stage, I think this paper requires a deep revision to emphasize the novelty of this study. Following questions must be addressed for better presentation
(1) The principle of the FBA should be illustrated schematically; it is critical for providing a background of this paper.
(2) The framework of the MPC seems to be general, and there is nothing new in modelling an FBA with the most general form of the state space, the presentation might be revised for a better introduction of the problem.
(3) The experimental part seems to be OK, and the conclusion might involve some quantitative information.
In addition, the format of this manuscript should be adapted and some spelling mistakes needs to be corrected carefully, e.g., “force-blance” in the abstract.
It seems to be OK.
Reviewer 3 Report
1. Abbreviations such as MPC and LQR must be explained.
2. How is Eq. (1) derived? A free-body diagram needs to be inserted.
3. What is F(t)? Authors state " F(t) is the inertial force excitation vector due to the input of acceleration.
3. Is U2 in Eq. (4) U2(t-\tau)?
4. What is l in Eq. (6)?
5. How is the control law, Eq. (21) implemented in real time?
6. What are two cases for Eqs. (g) and (h)?
7. If authors developed a new accelerometer, then the output of the new accelerometer should be compared to the output of other accurate accelerometer. Authors compared the measured signal with the input signal. It is difficult to understand authors' approach.
8. How was the control force generated? What was measured and what was applied to the sensor? How was the control algorithm implemented by the FPGA?
9. Authors proposed the LQR control strategy in [25]. Compared to the result of [25], what is the improvement of the current study?
10. Overall, the paper is poorly structured and written in a way that makes it difficult to understand the mathematical derivation process. It should be reorganized to make it easier for readers to understand.
N/A
Round 2
Reviewer 2 Report
it seems to be OK.
it seems to be OK.
Reviewer 3 Report
This paper is revised according to reviewer's comments.
English is OK.